# Salivary Molecular Testing for Periodontal Pathogen Monitoring: Clinical Performance of Flexible RT-PCR Platforms in Preventive Care Settings

**DOI:** 10.3390/diagnostics15192548

**Published:** 2025-10-09

**Authors:** Fabiana D’Urso, Federica Paladini, Mauro Pollini, Francesco Broccolo

**Affiliations:** Department of Experimental Medicine (DiMeS), University of Salento, 73100 Lecce, Italy; fabiana.durso@unisalento.it (F.D.); federica.paladini@unisalento.it (F.P.); mauro.pollini@unisalento.it (M.P.)

**Keywords:** periodontitis, saliva, molecular platforms, bacterial load, periodontal pathogens, probiotics, prevention

## Abstract

**Objective:** This study aimed to validate the clinical utility of a salivary molecular platform (Oral Predict^®^) for periodontal pathogen detection across preventive, therapeutic, and maintenance settings. **Methods:** A longitudinal randomized study was conducted involving 78 adults who provided saliva samples at baseline, one month, and three months after professional dental hygiene. Participants were randomized into two groups: control group (n = 39) and probiotic group with Oral Predict^®^ probiotic supplementation (n = 39). Crude saliva was processed directly without nucleic acid extraction and analyzed by multiplex real-time PCR using either the compact Real-time PCR system or standard thermocyclers. **Results:** At baseline, *Fusobacterium nucleatum* was the most prevalent pathogen (84.6%), followed by *Tannerella forsythia* (53.8%) and *Porphyromonas* gingivalis (46.2%). The Total Pathogen Burden Score (TPBS) showed progressive increases with age, smoking, and poor oral hygiene, and was significantly higher in participants with gingival bleeding. Among individual pathogens, no significant associations were observed with periodontitis staging or grading. Professional hygiene induced mean reductions of 1–2 logs across all pathogens, with TPBS decreasing from 8.7 ± 3.2 to 4.1 ± 2.8 (*p* < 0.001). At three months, 69.2% of the control group experienced bacterial rebound, whereas 85% of probiotic users sustained or improved bacterial reductions. **Conclusions**: Salivary molecular testing provides a robust, non-invasive approach for periodontal pathogen detection, treatment monitoring, and long-term maintenance assessment. The flexibility of the Oral Predict^®^ platform across point-of-care and laboratory settings, combined with automated interpretation, supports integration into preventive protocols and personalized periodontal care. These findings demonstrate the potential of saliva-based molecular diagnostics to shift periodontal management from reactive to predictive and precision-based strategies.

## 1. Introduction

Periodontitis is a major global health burden affecting nearly 800 million people [1]. It is associated with systemic diseases such as type 2 diabetes, cardiovascular disorders, and neurodegeneration [2,3,4,5]. Traditional diagnostic methods rely on invasive probing and radiography, which detect disease only after tissue destruction [6,7]. Saliva has emerged as a non-invasive alternative capable of capturing the whole oral microbiome [8,9,10]. Unlike subgingival plaque sampling (paper points, curettes, or probes), saliva collection is painless, quick, and suitable for repeated monitoring [10]. This approach enables early detection of pathogenic bacteria before localized tissue destruction becomes clinically apparent, shifting the paradigm from reactive treatment to preventive intervention [11]. The non-invasive nature of saliva collection eliminates patient discomfort while enabling frequent monitoring that can track treatment response and detect early signs of disease recurrence [11]. The clinical utility of salivary molecular testing extends beyond disease detection to include monitoring bacterial load changes in response to different therapeutic interventions, enabling clinicians to optimize treatment protocols based on objective molecular data rather than clinical symptoms alone. The integration of molecular diagnostic technologies with flexible analytical platforms represents a transformative development that brings sophisticated laboratory capabilities directly to clinical practice while adapting to different throughput requirements [12].

This study evaluates the Oral Predict^®^ saliva test, a quantitative multiplex PCR assay validated on flexible platforms (bCUBE^®^ or 96-well thermocyclers), for its clinical utility in pathogen detection, correlation with risk factors, and treatment monitoring.

## 2. Materials and Methods

### 2.1. Study Design and Participants

A randomized longitudinal study (June–December 2024) enrolled 78 adults aged 30–55 years. Exclusion criteria included antibiotics < 3 months, pregnancy, immunocompromise, periodontal therapy < 6 months, and medications inducing xerostomia (antihypertensives, antidepressants, antihistamines). Ethics approval was obtained (Univ. Salento Protocol 2024/015, 12 September 2025; Palermo 1 Protocol 8, 14 September 2022). The study was conducted according to the Declaration of Helsinki [13]. Seventy-eight participants were recruited from the DentalPro Clinic in Milan.

### 2.2. Study Timeline and Randomization

The study design incorporated three time points: baseline (T0), one month post-treatment (T1), and three months post-treatment (T2). Following baseline assessment, all participants received standardized professional dental hygiene including ultrasonic scaling, hand instrumentation, and polishing. Participants were randomized using computer-generated sequences into a control group (n = 39) receiving standard oral hygiene instructions, or probiotic group (n = 39) receiving standard care plus daily probiotic supplementation.

### 2.3. Probiotic Intervention

The probiotic group received standard care plus daily Oral Predict^®^ probiotic supplementation containing *Lactobacillus salivarius* LS97, *Lactobacillus plantarum* Lp05, *Lactobacillus paracasei* LC86, and *Streptococcus salivarius* ST81, each at 2.5 × 10^9^ CFU.

Participants consumed a total of 10^10^ CFU daily for three months, dissolving contents in the mouth after evening oral hygiene.

### 2.4. Clinical Assessment

Comprehensive clinical data collection included demographic information, smoking status, systemic health conditions, and oral health parameters. Clinical assessments were performed by two calibrated examiners with inter-examiner reliability assessed using Cohen’s kappa coefficient (κ = 0.89 for probing depth measurements). Intra-examiner reliability was evaluated through duplicate measurements on 10% of participants (κ = 0.92). Oral hygiene was assessed using the Simplified Oral Hygiene Index with categorization as good (0–1.2), intermediate (1.3–3.0), or poor (>3.0). Gingival bleeding was evaluated using the Bleeding on Probing index [13,14]. Smokers were included (46% of participants) rather than excluded because smoking is a major risk factor for periodontal disease, and our study aimed to evaluate the molecular platform’s ability to detect bacterial load variations across different risk profiles, including this important clinical population.

### 2.5. Salivary Sample Collection and Processing Workflow

Unstimulated whole crude saliva samples were collected following standardized protocols optimized for molecular analysis [14]. Participants avoided eating, drinking, or oral hygiene for two hours before collection, avoided alcohol and caffeine for 12 h, and rinsed with water 30 min prior to sampling. The passive drooling technique collected 2–3 mL saliva into sterile ESAliva collection tubes (ESAmed, Padova (PD), Italy) within 5 min and stored immediately at 4 °C. The study implemented a streamlined workflow designed to maximize efficiency while minimizing processing time and costs (Figure 1). Upon arrival at the laboratory, unprocessed saliva samples (without centrifugation) were analyzed using direct mutiplex quantitative PCR (qPCR), eliminating the need for nucleic acid extraction [15].

### 2.6. Molecular Analysis Platform

The Oral Predict^®^ saliva test detects and quantifies seven key periodontal pathogens (*Porphyromonas gingivalis*, *Tannerella forsythia*, *Treponema denticola*, *Prevotella intermedia*, *Fusobacterium nucleatum*, *Aggregatibacter actinomycetemcomitans*, and *Parvimonas micra*) by multiplex qPCR utilizing primers and TaqMan probe previously described [16,17,18].

#### Platform Flexibility

The Oral Predict^®^ test provides two analytical platforms tailored to laboratory throughput needs. The bCUBE^®^ System (Hyris Ltd., London, UK) is a compact device for small-scale applications (1–16 samples), using dual-channel fluorescence detection (FAM and HEX) with a 90-min turnaround, making it suitable for point-of-care testing and immediate clinical decision-making [19] (Figure 2A). For higher throughput, the CFX-96 format (Bio-Rad, Hercules, CA, USA) allows analysis of up to 96 samples on standard PCR plates with multi-channel fluorescence detection (FAM, HEX, Texas Red, Cy5). Laboratories can therefore choose the bCUBE^®^ for rapid, small-volume testing or the CFX-96 for centralized batch processing.

An integrated artificial intelligence algorithm automatically interprets Ct values and generates clinically relevant bacterial load classifications. A valid analytical session requires all positive controls to show amplification curves and all negative controls to show no signal. If these criteria are not met, samples are labeled as “Invalid” and must be retested. For valid samples, each pathogen is categorized as “Absent,” “Low Load,” “Medium Load,” or “High Load” based on Ct values. Classification thresholds for bacterial load categories were established based on Oral Predict^®^ saliva test with pathogen-specific cut-offs: for *P. gingivalis* and *A. actinomycetemcomitans* (high < 20, medium 20–23, low > 23 Ct values), *T. forsythia* (high < 21, medium 21–24, low > 24), *T. denticola* and *P. micra* (high < 19, medium 19–22, low > 22 Ct values), *P. intermedia* (high < 18, medium 18–21, low > 21 Ct values), *F. nucleatum* (high < 17, medium 17–20, low > 20 Ct values), and similarly defined ranges for each pathogen based on their individual association with periodontal disease risk as specified in the manufacturer’s protocol. The bCUBE^®^ AI algorithm automatically processes these classifications, generating standardized reports that translate molecular data into actionable clinical information without requiring specialized expertise in molecular diagnostics interpretation.

The top image displays the instrument with the reaction plate loading area (Figure 2A) and the schematic representation of the 16-well reaction plate layout for the Oral Predict^®^ test (Figure 2B). The plate is organized into four columns: Sample1 (blue wells), Sample2 (yellow wells), Positive Controls (red wells), and Negative Controls (green wells). Each column contains four master mix wells (Master mix 1–4) corresponding to different pathogen detection panels. This configuration allows simultaneous testing of two samples with appropriate quality controls in a single analytical session (Figure 2B).

### 2.7. Statistical Analysis

Sample size calculation was based on detecting a 1 log bacterial reduction with 80% power at α = 0.05. Descriptive statistics characterized population demographics. Spearman’s correlation assessed relationships between bacterial loads and clinical variables. Mann-Whitney U tests compared two groups, Kruskal-Wallis tests compared multiple groups. Longitudinal analysis used mixed-effects models accounting for repeated measurements. Log reduction calculations quantified treatment efficacy. Multiple testing correction employed the Benjamini–Hochberg procedure. Analysis used R software version 4.3.0 with significance at *p* < 0.05.

## 3. Results

### 3.1. Study Population Characteristics

The study population of 78 participants demonstrated substantial demographic diversity with a mean age of 48.7 ± 16.2 years (range 28–69). Baseline demographic characteristics showed no significant differences between control and probiotic groups for age (*p* = 0.752), smoking rate (*p* = 0.68), or oral hygiene scores (*p* = 0.59), ensuring valid comparisons. Table 1 shows the detailed demographic characteristics. Smoking status revealed 36 current smokers (46.2%) and 42 non-smokers (53.8%). Oral hygiene assessment showed 45 participants with good hygiene (57.7%), 24 with intermediate (30.8%), and 9 with poor hygiene (11.5%). Twelve participants had diabetes mellitus (15.4%) and 24 presented with gingival bleeding (30.8%).

### 3.2. Baseline Bacterial Prevalence and Load Distribution

Oral Predict^®^ saliva testing demonstrated clinical utility for bacterial load assessment across the study population. At baseline, bacterial prevalence analysis revealed heterogeneous colonization patterns across the study population. *F. nucleatum* emerged as the most prevalent pathogen, being present in over four out of five participants, predominantly at low-to-medium loads. *T. forsythia* was detected in just over half of the cohort, again largely at lower concentrations. Both *P. gingivalis* and *P. intermedia* exhibited identical prevalence rates, affecting nearly half of the subjects, while *T. denticola* and *P. micra* were identified in approximately one third of the cases, with very few individuals harboring high bacterial loads. *A. actinomycetemcomitans* demonstrated the lowest prevalence overall, being confined to less than one quarter of the population, and consistently detected at low-to-moderate levels (Table 2).

### 3.3. Total Pathogen Burden Score Analysis

Total Pathogen Burden Score (TPBS) calculation using high load = 3 points, medium load = 2 points, low load = 1 point revealed important distribution patterns. Nine patients (11.5%) showed no pathogen burden (TPBS = 0), 54 patients (69.2%) demonstrated low burden (TPBS 1–5), and 15 patients (19.2%) exhibited moderate burden (TPBS 6–10). No patients demonstrated high (11–15) or severe burden (16–21) scores. Red Complex analysis (*P. gingivalis*, *T. forsythia*, *T. denticola*) showed 24 patients negative (30.8%), 42 with low burden 1–3 points (53.8%), 12 with moderate burden 4–6 points (15.4%), and none with high burden 7–9 points. Orange Complex assessment (*P. intermedia*, *F. nucleatum*, *P. micra*) revealed 6 patients negative (7.7%), 57 with low burden (73.1%), 15 with moderate burden (19.2%), and none with high burden (Table 3).

### 3.4. Clinical Correlations

The molecular platform successfully detected correlations between bacterial load and established risk factors. Age showed significant positive correlations with bacterial loads. *F. nucleatum* demonstrated the strongest correlation (ρ = 0.623, 95% CI: 0.421–0.825, *p* = 0.002), followed by *T. forsythia* (ρ = 0.456, *p* = 0.018) and *P. gingivalis* (ρ = 0.342, *p* = 0.045). Total burden score showed moderate correlation with age (ρ = 0.542, *p* = 0.004). Age-stratified analysis revealed mean TPBS of 3.2 ± 2.1 for young adults, 5.8 ± 3.2 for middle-aged, and 6.4 ± 2.8 for older participants. Smoking showed dramatic associations with pathogen presence. *P. gingivalis* prevalence was 66.7% in smokers versus 28.6% in non-smokers (OR = 5.00, 95% CI: 1.23–20.3, *p* = 0.025). *T. forsythia* showed 75.0% prevalence in smokers versus 35.7% in non-smokers (OR = 5.40, *p* = 0.031). Smokers demonstrated significantly higher mean TPBS (7.2 ± 3.1) compared to non-smokers (4.1 ± 2.2, *p* = 0.019). Oral hygiene levels showed strong inverse relationships with bacterial loads using Kruskal-Wallis testing. *P. gingivalis* mean scores were 0.8 ± 1.1 for good hygiene, 1.9 ± 1.2 for intermediate, and 2.4 ± 0.9 for poor hygiene (H = 12.45, *p* = 0.002). Molecular-clinical correlation analysis using the established cut-offs revealed significant associations between bacterial load stratification and periodontal parameters. Molecular–clinical correlation analyses, based on pathogen-specific Ct cut-offs, demonstrated a significant relationship between bacterial load stratification and periodontal parameters. In particular, the Total Pathogen Burden Score (TPBS) was higher in participants with gingival bleeding compared to those without and progressively increased across periodontitis stages. Among individual pathogens no significant associations were observed with periodontitis staging or grading.

### 3.5. Longitudinal Treatment Response

Professional dental hygiene achieved significant bacterial load reductions at T1 across all pathogens as shown in Figure 3.

### 3.6. Probiotic Intervention Outcomes

Bacterial load monitoring revealed distinct patterns between treatment groups, demonstrating the platform’s utility for maintenance therapy assessment. At the three-month follow-up (T2), bacterial load monitoring revealed clear differences between the treatment groups. As shown in Figure 4, the control group exhibited a rebound in the Total Pathogen Burden Score (TPBS), while the probiotic group maintained a sustained reduction with continued improvement over time. These divergent trajectories highlight the role of probiotic supplementation in preventing bacterial regrowth after professional dental hygiene. Pathogen-specific analysis further confirmed these findings (Figure 5). Between T1 (1 month) and T2 (3 months), the control group demonstrated universal bacterial rebound, with increases across all pathogens (*P. gingivalis* +0.8, *T. forsythia* +0.6, *T. denticola* +0.9, *P. intermedia* +0.7, *F. nucleatum* +0.5, *A. actinomycetemcomitans* +0.4, *P. micra* +0.6). In contrast, the probiotic group maintained bacterial suppression, showing consistent reductions (*P. gingivalis* −0.4, *T. forsythia* −0.6, *T. denticola* −0.3, *P. intermedia* −0.8, *F. nucleatum* −0.5, *A. actinomycetemcomitans* −0.2, *P. micra* −0.7).

Individual response patterns showed 69.2% of control participants experienced bacterial load increases from T1 to T2, while 84.6% of probiotic users showed continued bacterial reduction. Repeated measures ANOVA revealed significant time effect (F(2, 152) = 89.4, *p* < 0.001), group effect (F(1, 76) = 12.8, *p* = 0.002), and time × group interaction (F(2, 152) = 28.6, *p* < 0.001)). At three-month follow-up (T2), significant differences emerged between groups The control group experienced bacterial rebound while the probiotic group maintained or continued bacterial reduction.

The Table 4 shows the Total Pathogen Burden Score changes over time for both groups, demonstrating the sustained benefit of probiotic intervention. The table illustrates the critical difference in long-term outcomes, with the control group showing significant bacterial rebound (from 4.2 to 6.8) while the probiotic group continues improvement (from 4.0 to 2.9), resulting in a statistically significant difference at T2.

## 4. Discussion

The primary objective of this study was to clinically validate the Oral Predict^®^ salivary test as a tool for bacterial load monitoring. The platform proved useful in several clinical contexts: (1) preventive screening to identify individuals at risk before symptoms develop, (2) objective evaluation of treatment efficacy after professional hygiene, and (3) monitoring of maintenance therapy to support long-term bacterial control. This longitudinal investigation confirms the value of flexible salivary molecular platforms in periodontal care, demonstrating their relevance for prevention, treatment assessment, and long-term management.

The Oral Predict^®^ system offers a scalable solution that adapts to different clinical needs. The bCUBE^®^ platform provides rapid, point-of-care testing with AI-assisted interpretation, suitable for small practices, while the CFX-96 format allows high-throughput batch testing for larger laboratories. Both approaches guarantee analytical accuracy and clinical relevance. AI-assisted interpretation represents a major advance, allowing general dental practitioners to integrate molecular diagnostics into daily workflows without requiring specialized expertise [19]. Similarly, the ESAmed workflow improves efficiency by eliminating centrifugation, reducing costs and turnaround time without affecting assay performance [15].

Salivary diagnostics provide a unique advantage because sampling is non-invasive and repeatable. This makes it possible to perform regular bacterial load assessments, detect pathogens before clinical damage occurs, and move from reactive treatment to preventive, precision-based care [20].

Baseline prevalence data highlighted important features of pathogen distribution. *F. nucleatum*, present in 84.6% of participants, confirmed its role as a bridge species in periodontal disease. Demographic analyses showed applicability across diverse populations, while TPBS distribution indicated that most participants carried manageable bacterial loads, ideal for preventive intervention.

Professional dental hygiene produced significant bacterial reductions, averaging 1–2 logs across all pathogens, with a mean TPBS decrease of 4.6 points—a clinically relevant shift from moderate to low burden [21,22,23]. These findings validate the platform’s capacity for objective treatment monitoring. However, the bacterial rebound observed in 69.2% of controls between one and three months illustrates the limits of mechanical therapy alone and underlines the importance of molecular follow-up.

The probiotic arm further demonstrated the utility of molecular monitoring in maintenance therapy. Sustained bacterial reduction was observed in 85% of probiotic users compared with only 30.8% of controls. This supports probiotics as an effective long-term adjunct, with activity driven by complementary mechanisms such as competitive exclusion, antimicrobial production, immune modulation, and biofilm disruption [20,24,25,26]. Importantly, the platform was able to capture these effects across all tested pathogens, highlighting its broader ecological relevance.

Strong correlations with known risk factors reinforce the test’s clinical significance. Age (ρ = 0.623 for *F. nucleatum*), smoking (OR = 5.00 for *P. gingivalis*), and oral hygiene all showed consistent associations, supporting integration of molecular data into personalized preventive strategies. Furthermore, composite TPBS values revealed clearer patterns, being higher in patients with bleeding and progressively increasing across stages, suggesting its potential as a global marker of disease severity. These results provide objective confirmation of epidemiological evidence and move risk assessment beyond subjective observation. Overall, while single-pathogen associations were limited, integrated quantification of multiple bacteria produced a more consistent signal, strengthening the case for saliva-based molecular testing in risk stratification and longitudinal monitoring.

We recognize some limitations: microbial composition differs between gingivitis and periodontitis, and the lack of systematic periodontal staging in this study may have influenced some correlations. Future research should include standardized staging to better stratify microbial patterns by disease severity. In summary, Oral Predict^®^ salivary testing proved to be a reliable diagnostic approach, demonstrating strong pathogen–risk factor correlations, objective monitoring of treatment efficacy, and robust evaluation of maintenance strategies. Baseline results confirmed the dominance of *F. nucleatum*, while TPBS consistently reflected bleeding and disease progression, supporting its role as a global marker and aligning with established models of dysbiosis in periodontal disease [6,7].

## 5. Conclusions

Oral Predict^®^ salivary testing enables predictive periodontal care through non-invasive detection, objective treatment monitoring, and personalized risk stratification. The flexible platform (bCUBE^®^ or CFX96) adapts to diverse clinical contexts, while AI-assisted interpretation facilitates routine use. Combining professional hygiene with probiotics and salivary monitoring supports long-term disease control.

The study validates the clinical utility of Oral Predict^®^ testing for enabling predictive periodontal care through bacterial load monitoring for early pathogen detection, objective treatment monitoring, and personalized risk assessment across different clinical settings. The flexible platform approach successfully addresses varying clinical needs while maintaining high analytical accuracy and providing actionable clinical information through automated interpretation systems. The combination of professional hygiene with targeted probiotic supplementation, monitored through Oral Predict^®^ testing, offers a comprehensive approach to periodontal disease management that addresses both immediate bacterial load reduction and long-term microbial ecology stabilization. This dual intervention strategy enables personalized treatment protocols based on objective molecular data rather than subjective clinical assessment alone. These findings support the integration of salivary Oral Predict^®^ testing into standard periodontal care protocols as a tool for bacterial load monitoring enabling early disease detection, treatment efficacy assessment, and personalized preventive intervention strategies that could significantly improve long-term periodontal health outcomes across diverse patient populations.

Future research should focus on expanding the pathogen panel to include emerging periodontal bacteria, investigating optimal probiotic strain combinations for different patient populations based on molecular monitoring data, and developing predictive algorithms that integrate molecular bacterial load data with clinical parameters for personalized treatment planning. The establishment of standardized bacterial load thresholds for different age groups and risk categories will further enhance the clinical utility of salivary molecular testing in routine periodontal practice for preventive care, treatment monitoring, and maintenance therapy optimization.

## Figures and Tables

**Figure 1 diagnostics-15-02548-f001:**
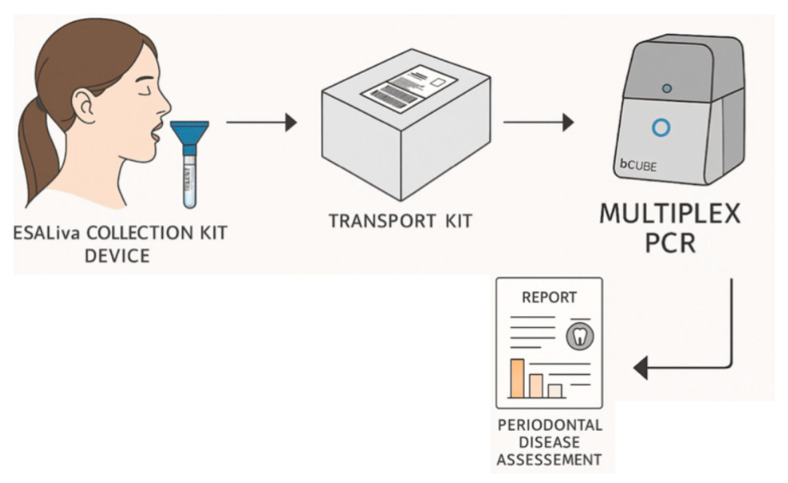
Salivary Periodontal Pathogen Detection Workflow. The workflow encompasses patient self-collection using the ESALiva collection device under dental professional supervision → sample stabilization in ESAmed preservation system → transport to laboratory → direct molecular processing without centrifugation → analysis using bCUBE^®^ (1–16 samples, AI-assisted interpretation) → automated report generation with periodontal disease assessment and bacterial load categorization.

**Figure 2 diagnostics-15-02548-f002:**
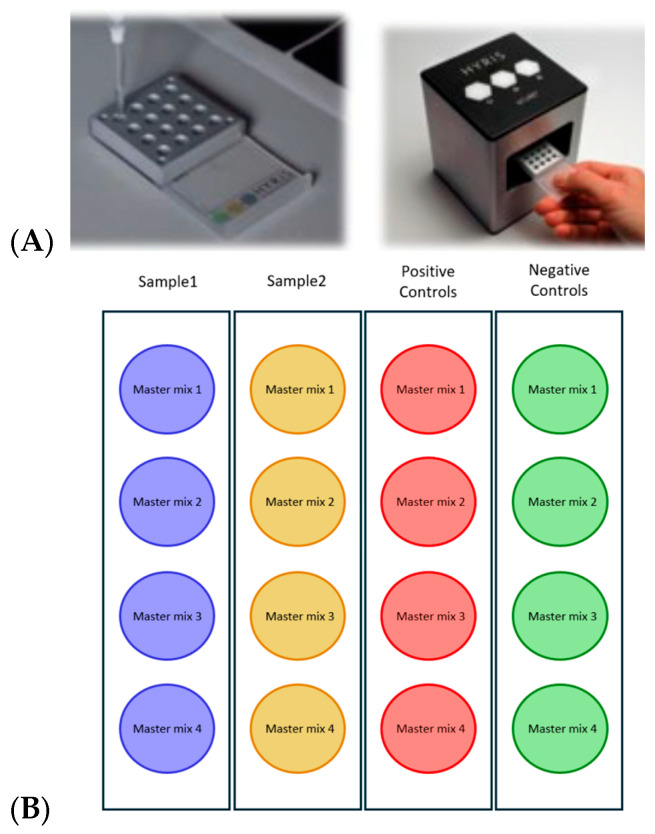
The Oral Predict^®^ saliva test can be analyzed using the compact bCUBE^®^ analytical system for small-scale applications (**A**) organized into four columns (**B**): Sample1 (blue wells), Sample2 (yellow wells), Positive Controls (red wells), and Negative Controls (green wells), each tested with fourdifferent master mixes.

**Figure 3 diagnostics-15-02548-f003:**
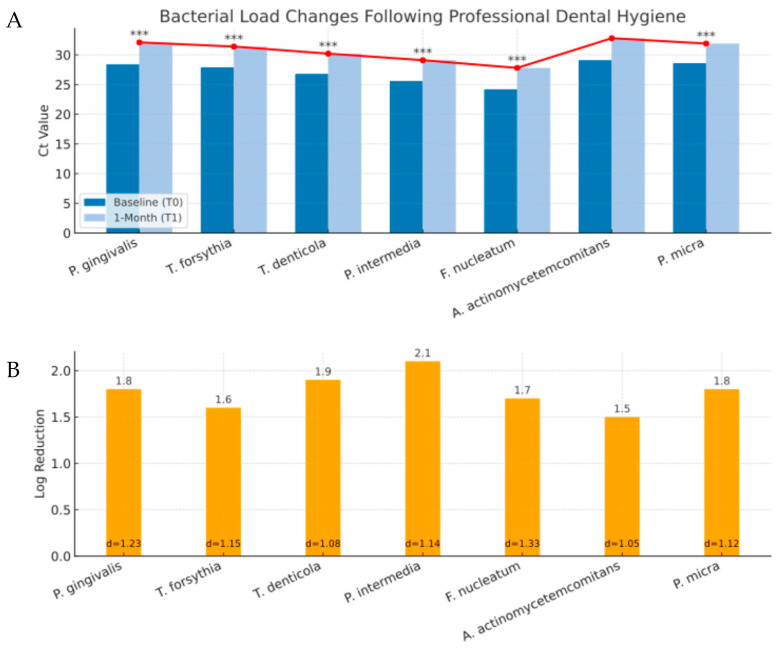
Bacterial load changes following professional dental hygiene. Bacterial load changes were measured in saliva samples before (T0) and one month after (T1) professional dental hygiene treatment. Mean Ct values for each bacterial species at T0 (dark blue) and T1 (light blue). The red line represents the trend in Ct value increase across bacterial species, indicating a reduction in bacterial load (since higher Ct corresponds to lower DNA quantity) (**A**). Logarithmic reduction in bacterial load (orange bars) between T0 and T1, with corresponding effect size (Cohen’s *d*) values shown below each bar (**B**). Mean log reductions were: *P. gingivalis* 1.8 ± 0.6 (*p* < 0.001), *T. forsythia* 1.6 ± 0.5 (*p* < 0.001), *T. denticola* 1.9 ± 0.7 (*p* < 0.001), *P. intermedia* 2.1 ± 0.8 (*p* < 0.001), *F. nucleatum* 1.7 ± 0.6 (*p* < 0.001), *A. actinomycetemcomitans* 1.5 ± 0.5 (*p* = 0.002), and *P. micra* 1.8 ± 0.6 (*p* < 0.001). Mean TPBS decreased from 8.7 ± 3.2 at baseline to 4.1 ± 2.8 at T1 (reduction 4.6 ± 1.9 points, *p* < 0.001) (**C**). Asterisks (*** indicate statistical significance: *p* < 0.001).

**Figure 4 diagnostics-15-02548-f004:**
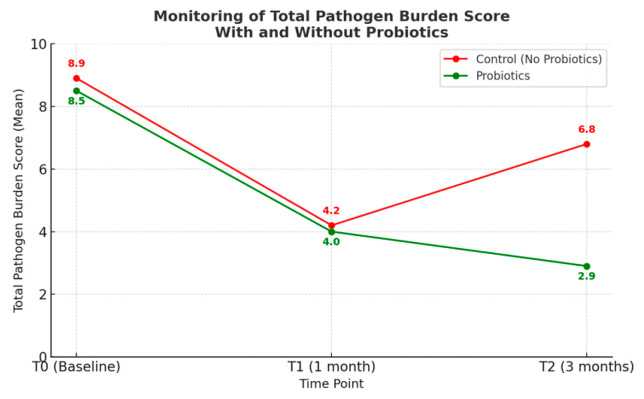
Monitoring of total pathogen burden score (TPBS) with and without probiotics. Line graph showing TPBS changes over time. Control group (red line): demonstrating bacterial rebound. Probiotic group (green line): showing sustained reduction with continued improvement.

**Figure 5 diagnostics-15-02548-f005:**
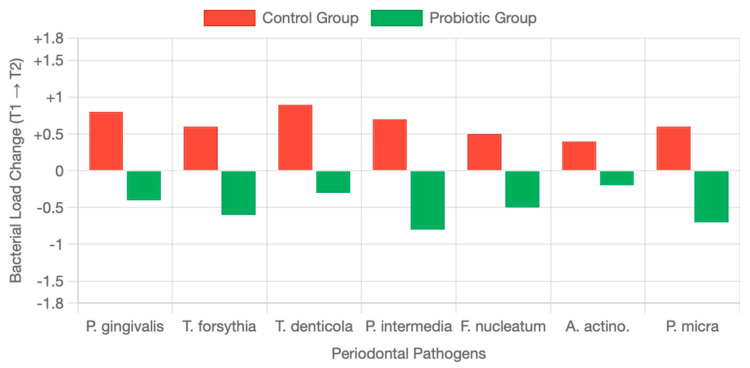
Change in bacterial load for Each Pathogen With and Without Probiotics from T1 to T2 (1 − month to 3 − month follow-up). Control group (red bars) shows increases: *P. gingivalis* +0.8, *T. forsythia* +0.6, *T. denticola* +0.9, *P. intermedia* +0.7, *F. nucleatum* +0.5, *A. actinomycetemcomitans* +0.4, *P. micra* +0.6. Probiotic group (green bars) shows decreases: *P. gingivalis* −0.4, *T. forsythia* −0.6, *T. denticola* −0.3, *P. intermedia* −0.8, *F. nucleatum* −0.5, *A. actinomycetemcomitans* −0.2, *P. micra* −0.7.

**Table 1 diagnostics-15-02548-t001:** Study Population Demographics and Clinical Characteristics.

Characteristic	n	%
Age (years)	78	
≤35 years	21	26.9
36–55 years	15	19.3
>55 years	42	53.8
Gender		
Female	54	69.2
Male	24	30.8
Smoking Status		
Current smokers	36	46.2
Non-smokers	42	53.8
Oral Hygiene Level		
Good (0–1.2)	45	57.7
Intermediate (1.3–3.0)	24	30.8
Poor (>3.0)	9	11.5
Diabetes Mellitus	12	15.4
Gingival Bleeding	24	30.8
Periodontal diagnosis		
Gingivitis	31	39.7
Staging of periodontitis		
I	9	11.5
II	6	7.7
III	16	20.5
IV	16	20.5
Grading of periodontitis		
A	30	38.5
B	9	11.5
C	4	5

**Table 2 diagnostics-15-02548-t002:** Baseline Bacterial Prevalence and Load Distribution.

Pathogen	Total Positive n (%)	High Load n (%)	Medium Load n (%)	Low Load n (%)
*F. nucleatum*	66 (84.6%)	6 (7.7%)	24 (30.8%)	36 (46.2%)
*T. forsythia*	42 (53.8%)	3 (3.8%)	12 (15.4%)	27 (34.6%)
*P. gingivalis*	36 (46.2%)	6 (7.7%)	9 (11.5%)	21 (26.9%)
*P. intermedia*	36 (46.2%)	3 (3.8%)	12 (15.4%)	21 (26.9%)
*P. micra*	30 (38.5%)	0 (0%)	9 (11.5%)	21 (26.9%)
*T. denticola*	27 (34.6%)	3 (3.8%)	6 (7.7%)	18 (23.1%)
*A. actinomycetemcomitans*	18 (23.1%)	0 (0%)	6 (7.7%)	12 (15.4%)

**Table 3 diagnostics-15-02548-t003:** Total Pathogen Burden Score Distribution.

Burden Category	Score Range	Count	Percentage
No Burden	0	9	11.5%
Low Burden	1–5	54	69.3%
Moderate Burden	6–10	15	19.2%
High Burden	11–15	0	0%
Severe Burden	16–21	0	0%

**Table 4 diagnostics-15-02548-t004:** Total Pathogen Burden Score Changes Over Time.

Time Point	Control Group	Probiotic Group	*p*-Value
T0 (Baseline)	8.9 ± 3.4	8.5 ± 3.0	0.752
T1 (1 month)	4.2 ± 2.9	4.0 ± 2.7	0.863
T2 (3 months)	6.8 ± 2.9	2.9 ± 2.1	<0.001

## Data Availability

All data are available on request from the corresponding author.

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
