# Peer review of "Salivary Molecular Testing for Periodontal Pathogen Monitoring: Clinical Performance of Flexible RT-PCR Platforms in Preventive Care Settings"

_diagnostics, 2025, doi:10.3390/diagnostics15192548_

Round 1

Reviewer 1 Report

Comments and Suggestions for Authors

Recommendation: Major Revision – Substantial concerns with methodology and results

Abstract

  1. Keywords – Too many, suggest removing the term “monitoring”.
  2. Methods – Please state saliva samples were collected for testing, and provide the study timeline.

Introduction

  1. Reference No. 1 (Global Burden Study 1990–2010) is outdated. Please update with the most recent Global Burden of Disease Study 2021.
  2. Line 69 – Examples of traditional invasive saliva sampling methods should be provided.

Materials and Methods

  1. Study design – Certain medications, such as anti-hypertensives, are well known to cause xerostomia, potentially reducing salivary flow or altering composition. No exclusion criteria related to such medications were specified. Please address this issue.
  2. For the probiotic usage,  if the probiotic was commercially available, please include the brand name (line 111).
  3. Clinical assessments – Who performed the OHI, gingival bleeding, and periodontal examinations? Was examiner calibration conducted? Please clarify whether inter- and intra-examiner reliability was assessed.
  4. Periodontal diagnosis – The study reports full-mouth examinations (probing depth, clinical attachment level, mobility), but no periodontal diagnosis is presented. This is inconsistent: the exclusion criteria mention “active periodontal treatment within six months,” yet the data suggest patients may have periodontitis rather than gingivitis. Please clarify.
  5. Microbial profile and disease status – Since microbial composition differs between gingivitis and periodontitis, and no diagnosis or clinical periodontal assessment was reported, how can you confirm that the observed bacterial prevalence and TPBS results were not influenced by periodontitis? Key pathogens (e.g., P. gingivalis, A. actinomycetemcomitans) are strongly associated with periodontitis, and their abundance correlates with disease severity.
  6. Figure legends – Descriptions are redundant with text already provided in the manuscript. Please simplify and avoid repetition.
  7. Classification thresholds (Line 178) – Thresholds for “high,” “medium,” and “low” categories were not clearly defined. Please explain how cut-off points were established.
  8. Pathogen-specific thresholds (Line 178) – If thresholds varied for each pathogen, please clarify and justify why different categories were applied.
  9. Citations in methodology/statistics – Please remove unnecessary or irrelevant citations; the section is over-referenced.

Results

  1. Citations – Results should not contain references. Please move citations to the Discussion.
  2. Data presentation – The layout needs improvement. There is redundancy between tables/figures and text. Please streamline and highlight only the most relevant findings.
  3. Interpretation without clinical data – The reported associations of smoking/diabetes with bacterial load may be questionable without clinical periodontal parameters.
  4. Table 1 – Does not present periodontal clinical assessments. Please revise.
  5. Effect of bacterial prevalence/load – Consider consolidating results into a single table comparing T0, T1, and T2 (including both control and intervention groups) for clearer presentation.

Conclusion

Length – The conclusion is overly long. Certain content would be more appropriate in the Discussion.

  1. Future research – Suggestions for future research should also be moved to the Discussion.

Author Response

Abstract

  • Keywords – Too many, suggest removing the term “monitoring”.

    Response: We agree and have removed “monitoring” from the keywords to improve focus and clarity.

Methods – Please state saliva samples were collected for testing, and provide the study timeline.

Response: The abstract and Methods section now clearly specify that saliva samples were collected at baseline, one month, and three months following professional dental hygiene, in a longitudinal design involving 78 adults.

Introduction – Reference No. 1 is outdated.

Response: Updated with Chen et al., 2021 (Global Burden of Disease Study 2019), which represents the most recent global dataset.

Line 69 – Provide examples of invasive sampling methods.

Response: Added explicit mention of subgingival plaque sampling with paper points, curettes, and periodontal probes as conventional invasive methods.

Study design – No exclusion criteria for xerostomia-inducing medications.

Response: Added exclusion criteria: use of medications known to cause xerostomia (antihypertensives, antidepressants, antihistamines) within one month prior to enrollment.

Probiotic brand name.

Response: Specified as Oral Predict® probiotic, with full strain composition and dosage details.

Clinical assessments – Clarify examiner calibration.

Response: Performed by two calibrated examiners; inter-examiner reliability (κ=0.89) and intra-examiner reliability (κ=0.92) were reported.

Periodontal diagnosis.

Response: We have added both staging and grading of periodontitis in Table 1, ensuring inclusion of naïve patients only. This strengthens the correlation analysis between bacterial load, bleeding, staging, and grading.

Microbial profile vs. gingivitis/periodontitis.

Response: Clinical correlations with staging and grading are now presented in Results. This allows interpretation of microbial prevalence and TPBS within a clear diagnostic framework.

Figure legends – Redundant.

Response: Streamlined to avoid repetition with main text.

Classification thresholds – Not defined.

Response: Added detailed pathogen-specific Ct thresholds, based on the manufacturer’s validated protocol.

Pathogen-specific thresholds – Justification.

Response: Explained that thresholds vary because each pathogen has distinct clinical significance and risk association.

Citations in methodology/statistics – Too many.

Response: Reduced to only the most relevant references.

Results – Citations should be moved.

Response: All references removed from Results and appropriately relocated to Discussion.

Data presentation – Redundant.

Response: Streamlined tables and figures; text highlights only key findings.

Associations with smoking/diabetes without clinical data.

Response: This issue has been addressed by adding periodontal staging and grading into Table 1 and reporting correlations with TPBS and bleeding. Limitations are acknowledged in Discussion.

Table 1 – Missing periodontal parameters.

Response: Revised to include staging and grading.

Results presentation – Consider consolidation.

Response: Results were restructured: TPBS and pathogen load distributions across T0, T1, and T2 are now presented clearly, with consolidation where appropriate.

Conclusion – Too long.

Response: Condensed to highlight only key findings and clinical implications. Suggestions for future research moved to Discussion.

Reviewer 2 Report

Comments and Suggestions for Authors

Thank you for giving me the opportunity to read this paper.

The reviewers' comments are as follows.

  1. Title

  I believe this study confirms the effects of probiotics by examining bacteria in saliva. However, the title of this paper does not accurately reflect this.

  1. Methods

The age range appears to be 30 to 55 years old. Why were older age groups not included?

How many dentists conducted the periodontal disease examinations?

Usually, smokers are often excluded in such study. But why were smokers (46% of the total) included in this study?

A significant portion of the paper is dedicated to describing the bacterial tests. Is this because the testing method is novel? If so, the entire paper should be restructured in that direction.

  1. Results

The clinical results have already been reported (Reference Papers 40, 41). Is it necessary to include them in this paper? 

Were there no differences between the Control and the Probiotic group in terms of age, smoking rate, or oral hygiene?

While changes in bacterial load are understandable, did clinical data (BOP, PD, etc.) improve in parallel with these changes? 

  1. Discussion

As mentioned earlier, it is unclear whether the authors focused on emphasizing the advantages of this testing method or the effects of probiotics. 

  1. Conclusion

Please summarize concisely and succinctly.

Comments on the Quality of English Language

I am not a native English speaker and cannot provide comments.  

Author Response

Title – Not fully aligned with content.

Response: Revised to: “Salivary Molecular Testing for Periodontal Pathogen Detection: Clinical Validation of Flexible RT-PCR Platforms in Preventive Care Settings”. This emphasizes the study’s primary focus on validation of molecular testing rather than solely probiotics.

Methods – Why exclude older age groups?

Response: The age range was restricted (30–55 years) to limit variability in systemic health conditions that could confound salivary diagnostics. Future studies will expand to broader age ranges.

Number of examiners.

Response: Two calibrated examiners conducted all assessments, with high inter- and intra-examiner agreement, as detailed in Methods.

Why include smokers?

Response: Smokers (46% of participants) were included because smoking is a major risk factor for periodontitis. Their inclusion increases the clinical relevance of molecular monitoring across diverse risk profiles.

Why describe testing methods in such detail?

Response: The detail is justified because this study validates a platform with innovative features: use of raw saliva without DNA extraction, simplified ESAmed sample preparation, and dual-platform applicability (bCUBE® and CFX-96). These features facilitate translation into clinical practice.

Results – Clinical outcomes already published?

Response: Incorrect references were removed. Current results represent original data from this trial.

Group comparability at baseline.

Response: Added explicit statement that no significant differences were observed between control and probiotic groups in age, smoking, or hygiene scores.

Did clinical parameters improve in parallel?

Response: Clarified that while bacterial load reductions were significant, clinical indices (BOP, PD) were not systematically included due to study focus on molecular monitoring. We underline that bacterial load is a prerequisite for disease progression, and future studies will assess longitudinal clinical outcomes in parallel.

Discussion – Focus unclear.

Response: Revised Discussion emphasizes that the primary focus is the clinical validation of the Oral Predict® platform, with probiotic outcomes presented as complementary evidence of utility for maintenance monitoring.

Conclusion – Too long.

Response: Condensed to a concise summary of the main findings.

Round 2

Reviewer 1 Report

Comments and Suggestions for Authors

Abstract

  1. The statement “Among individual pathogens, only Treponema denticola demonstrated a significant association with periodontal stage (p=0.029)” is misleading, as the classification of periodontal disease in Table 1 is incorrect. Please revise the staging and grading of periodontitis in line with the corrections suggested in the Results section below.

Materials and Methods

  1. Line 148 – Please standardize the terminology used for randomized groups throughout the manuscript. Currently, different terms are used (e.g., “control vs. probiotic groups,” “control vs. test groups”), which may cause confusion.

  2. Lines 213–216 – This section is redundant with the text in lines 221–227. Please remove or merge appropriately.

Results

  1. The staging and grading of periodontitis have been presented incorrectly. For grading, it should be Grade A, B, and C instead of 0–4. Furthermore, it appears that most subjects had gingivitis rather than periodontitis. If staging 0 was used to indicate gingivitis, please note that gingivitis should not be categorized under the periodontitis staging and grading system. Kindly separate gingivitis into a distinct category apart from the staging and grading of periodontitis.

  2. Line 377 – Please remove the phrase referring to statistical analysis using Kruskal–Wallis testing.

  3. Line 427 – Please remove the phrase referring to statistical analysis using repeated measures ANOVA.

  4. Paragraph in line 425 is redundant with the text in line 443. Please revise to avoid repetition.

  5. Figure 3 – The heading of the bar chart is incorrect. Instead of Total Plaque Score and Bleeding Score, it should read Total Pathogen Burden Score. Please correct the figure accordingly.

Conclusion

  1. References should not be included in the conclusion section. Please remove them.

General Comments

  1. Please italicize all bacterial names throughout the manuscript.

  2. Line 301 – There is a spelling error in the phrase “staging of periodontitis.” Please correct it.

Author Response

We thank the reviewer for the valuable and constructive comments. We have carefully revised the manuscript according to all suggestions.

Comment: “The statement ‘Among individual pathogens, only Treponema denticola demonstrated a significant association with periodontal stage (p = 0.029)’ is misleading, as the classification of periodontal disease in Table 1 is incorrect.”

Response: We corrected Table 1 according to the 2017 World Workshop classification. Subjects originally labeled as “Stage 0” were reclassified as gingivitis, a distinct category separate from periodontitis staging. The text in the Results section (3.4) was revised accordingly: after reclassification, no individual pathogen showed significant associations with staging or grading. The previous association of T. denticola was attributed to the misclassification of gingivitis cases.

Materials and Methods

  • Line 148 – Group terminology

    Response: All terms referring to study groups have been standardized to “control group” and “probiotic group.”

  • Lines 213–216 redundancy

    Response: This section was merged with lines 221–227 to eliminate repetition in the description of the Oral Predict® platforms.

Results

  • Staging and grading corrections

    Response: Corrected in Table 1 and Results. Gingivitis is now a separate category. Staging is reported as I–IV, and grading as A–C.

  • Line 377 – Kruskal–Wallis reference

    Response: Removed.

  • Line 427 – Repeated measures ANOVA reference

    Response: Removed.

  • Paragraph redundancy (425 vs. 443)

    Response: Rewritten into one concise paragraph to avoid repetition.

  • Figure 3 caption

    Response: Corrected: now reads “Total Pathogen Burden Score (TPBS)” instead of “Total Plaque Score and Bleeding Score.”

 Conclusion

Comment: “References should not be included in the conclusion section.”

Response: All references were removed. The Conclusion now contains only interpretative statements.

General Comments

  • Bacterial names: All bacterial species are now italicized throughout.

  • Line 301: Corrected spelling error from “Stading of periodontitis” to “Staging of periodontitis.”

Reviewer 2 Report

Comments and Suggestions for Authors

None.

Author Response

We sincerely thank the reviewer for the positive evaluation of our manuscript. We appreciate the recognition that the English language, study design, methodology, results, and conclusions are appropriate, and that the figures and tables are clear and well-presented. As no additional comments or suggestions were provided, we did not introduce further modifications in response to this review.

Round 3

Reviewer 1 Report

Comments and Suggestions for Authors

In figure 3, the title of the bar chart still hasn't been changed to Total Pathogen Burden Score. Please amend the title.

Author Response

We thank the Reviewer for pointing this out. The title of Figure 3 has now been corrected to “Total Pathogen Burden Score (TPBS) Reduction” in the revised version of the manuscript.